# $^{18}$F-AV-1451 and CSF T-tau and P-tau as biomarkers in Alzheimer's disease

Niklas Mattsson[1,2,3,*] (iD), Michael Schöll[1,4], Olof Strandberg[1], Ruben Smith[1,3], Sebastian Palmqvist[1,3], Philip S Insel[1,5,6], Douglas Hägerström[7], Tomas Ohlsson[8], Henrik Zetterberg[9,10,11], Jonas Jögi[12], Kaj Blennow[9,10] & Oskar Hansson[1,2,**] (iD)

## Abstract

To elucidate the relationship between cerebrospinal fluid (CSF) total-tau (T-tau) and phosphorylated tau (P-tau) with the tau PET ligand $^{18}$F-AV-1451 in Alzheimer's disease (AD), we examined 30 cognitively healthy elderly (15 with preclinical AD), 14 prodromal AD, and 39 AD dementia patients. CSF T-tau and P-tau were highly correlated ($R = 0.92$, $P < 0.001$), but they were only moderately associated with retention of $^{18}$F-AV-1451, and mainly in demented AD patients. $^{18}$F-AV-1451, but not CSF T-tau or P-tau, was strongly associated with atrophy and cognitive impairment. CSF tau was increased in preclinical AD, despite normal $^{18}$F-AV-1451 retention. However, not all dementia AD patients exhibited increased CSF tau, even though $^{18}$F-AV-1451 retention was always increased at this disease stage. We conclude that CSF T-tau and P-tau mainly behave as biomarkers of "disease state", since they appear to be increased in many cases of AD at all disease stages, already before the emergence of tau aggregates. In contrast, $^{18}$F-AV-1451 is a biomarker of "disease stage", since it is increased in clinical stages of the disease, and is associated with brain atrophy and cognitive decline.

**Keywords** Alzheimer; biomarker; cerebrospinal fluid; positron emission tomography; tau

**Subject Categories** Biomarkers & Diagnostic Imaging; Neuroscience

## Introduction

Alzheimer's disease (AD) is characterized by extracellular aggregation of amyloid β (Aβ) peptides and intracellular aggregation of phosphorylated tau proteins (Zetterberg & Mattsson, 2014). Aggregation of tau may be closely related to atrophy and cognitive decline in AD (Arriagada *et al*, 1992). Cerebrospinal fluid (CSF) measures of total-tau (T-tau) and phosphorylated tau (P-tau) may reflect axonal degeneration and tangle pathology in AD (Blennow *et al*, 2010). Recently, positron emission tomography (PET) tracers have been developed to visualize tau deposits *in vivo*. $^{18}$F-AV-1451 (formerly called $^{18}$F-T807; Chien *et al*, 2013; Xia *et al*, 2013) detects AD-type tau aggregates (Marquié *et al*, 2015; Smith *et al*, 2016a), differentiates AD patients from controls (Brier *et al*, 2016; Gordon *et al*, 2016; Johnson *et al*, 2016; Schöll *et al*, 2016; Schwarz *et al*, 2016) and correlates with regional changes in brain glucose metabolism in AD (Ossenkoppele *et al*, 2016; Smith *et al*, 2016b).

CSF and PET tau biomarkers correlate overall (Chhatwal *et al*, 2016; Gordon *et al*, 2016), but published results mainly comes from cognitively normal elderly, and includes few AD dementia cases. This has left several questions unresolved about the relationship between CSF and PET tau biomarkers in AD. It is not known if correlations between CSF and PET tau biomarkers vary across disease stages in AD. It is also unclear if CSF and PET tau biomarkers have different associations with neurodegeneration and cognition. Finally, it is unclear if dichotomous classifications of CSF and PET tau biomarkers produce concordant results. Here we address these questions using CSF T-tau, P-tau and $^{18}$F-AV-1451 in a cohort consisting cognitively healthy controls, and patients with prodromal AD and AD dementia.

1  Clinical Memory Research Unit, Faculty of Medicine, Lund University, Lund, Sweden
2  Memory Clinic, Skåne University Hospital, Lund, Sweden
3  Department of Neurology, Skåne University Hospital, Lund, Sweden
4  MedTech West and the Department of Psychiatry and Neurochemistry, University of Gothenburg, Gothenburg, Sweden
5  Department of Veterans Affairs Medical Center, Center for Imaging of Neurodegenerative Diseases, San Francisco, CA, USA
6  Department of Radiology and Biomedical Imaging, University of California, San Francisco, CA, USA
7  Department of Clinical Neurophysiology, Skåne University Hospital, Lund, Sweden
8  Department of Radiation physics, Skåne University Hospital, Lund, Sweden
9  Clinical Neurochemistry Laboratory, Sahlgrenska University Hospital, Mölndal, Sweden
10  Department of Molecular Neuroscience, Institute of Neuroscience and Physiology, The Sahlgrenska Academy at the University of Gothenburg, Mölndal, Sweden
11  Department of Molecular Neuroscience, UCL Institute of Neurology, London, UK
12  Department of Clinical Physiology and Nuclear Medicine, Skåne University Hospital, Lund, Sweden
   *Corresponding author. Tel: +46 (0)46 171000; E-mail: niklas.mattsson@med.lu.se
   **Corresponding author. Tel: +46 (0)40 331000; E-mail: oskar.hansson@med.lu.se

# Results

Demographics are presented in Table 1.

**CSF T-tau and P-tau are highly correlated**

We tested if CSF T-tau and P-tau were correlated. CSF T-tau and P-tau were highly correlated overall ($R = 0.92$, $P < 0.001$), and in controls ($R = 0.91$, $P < 0.001$), prodromal AD ($R = 0.74$, $P = 0.0025$), and AD dementia ($R = 0.94$, $P < 0.001$). The correlations were also seen when adjusting for covariates (Fig EV1).

**Correlations between CSF tau and [18]F-AV-1451 differ by diagnosis and region**

We next tested correlations between CSF tau biomarkers and regional [18]F-AV-1451 retention. The regions corresponded to different image-based stages of tau, as described by Cho *et al* (2016). These regions are labelled the tau stage I–II region (entorhinal cortex), the tau stage III region (parahippocampal gyrus, fusiform gyrus, amygdala), the tau stage IV region (inferior temporal and middle temporal gyri), the tau stage V region (posterior cingulate gyrus, caudal anterior cingulate gyrus, rostral anterior cingulate gyrus, precuneus, inferior parietal lobule, superior parietal lobule, insula, supramarginal gyrus, lingual gyrus, superior temporal gyrus, medial orbitofrontal gyrus, rostral middle frontal gyrus, lateral orbitofrontal gyrus, caudal middle frontal gyrus, superior frontal gyrus, lateral occipital gyrus) and the tau stage VI region (precentral gyrus, postcentral gyrus, paracentral gyrus). We also constructed a composite of tau stages I–V regions.

We used linear regression models with regional [18]F-AV-1451 as outcome and CSF T-tau or P-tau as predictors, adjusted for age, sex and diagnosis (Fig 1A–L and Table EV1). There were no significant correlations between CSF tau biomarkers and regional [18]F-AV-1451 in controls. In prodromal AD, CSF T-tau correlated with [18]F-AV-1451 in tau stage I–II regions (Fig 1A). In AD dementia, CSF T-tau

and P-tau correlated with [18]F-AV-1451 in tau stage III, V and VI regions, and in the stage I–V composite region (Fig 1B, D–F, H, J and L). The findings were similar when adjusting for atrophy (measured by cortical thickness in the temporal lobe, see the Materials and Methods section for details).

We also did voxel-wise regressions, which did not depend on any *a priori* defined regions. In these analyses, CSF T-tau and P-tau were significantly related to [18]F-AV-1451 retention predominantly in frontal and temporoparietal, as well as posterior medial temporal cortical areas (Fig 2A and C). This pattern generally agreed with the regional analyses. The voxel-wise associations were also present when adjusting for cortical atrophy (Fig 2B and D).

**[18]F-AV-1451 but not CSF tau is correlated with neurodegeneration**

We tested associations between tau biomarkers and brain structure. We used hippocampal volume and cortical thickness in the temporal lobe as outcomes, and CSF T-tau, P-tau and [18]F-AV-1451 in the stage I–V composite region as predictors. The analyses were adjusted for age, sex, diagnosis and (for hippocampal volume) intracranial volume.

Greater [18]F-AV-1451 retention was related to thinner cortex [$\beta = -0.23$ (unit: standardized uptake value ratio, SUVR), $P < 0.001$], but not to hippocampal volume ($P = 0.37$). CSF T-tau and P-tau were not associated with our atrophy measures ($P = 0.71$–$0.90$). The relationship between cortical thickness and [18]F-AV-1451 was not affected by adjusting for CSF tau biomarkers, but (surprisingly) higher levels of T-tau [$\beta = 0.00024$ (unit: ng/l), $P = 0.015$] and P-tau [$\beta = 0.0025$ (unit: ng/l), $P = 0.0021$] were associated with thicker cortex in these adjusted models.

Voxel-based morphometry (VBM) analyses of associations between tau biomarkers and atrophy confirmed that greater [18]F-AV-1451 was associated with thinner cortex, predominantly in temporoparietal areas (Fig 3A), and only showed very mild associations between CSF T-tau and P-tau with cortical thickness (Fig 3B and C). The associations between [18]F-AV-1451 and cortical thickness were largely unaffected by adjusting for CSF tau biomarkers (Fig EV2A and B), while the mild associations between CSF tau biomarkers and cortical thickness were further attenuated by adjusting for [18]F-AV-1451 (Fig EV2C and D).

**[18]F-AV-1451 but not CSF tau is correlated with worse cognition**

We tested associations between cognitive measures as outcomes and CSF T-tau, P-tau, [18]F-AV-1451, hippocampal volume and cortical thickness as predictors, adjusted for age, sex, education, diagnosis and (for hippocampal volume) intracranial volume (Fig 4A–C). Worse mini-mental state examination (MMSE) score was associated with thinner cortex ($P = 0.016$) and higher [18]F-AV-1451 retention ($P = 0.010$). Worse memory performance (ADAS-cog word list delayed recall) and worse attention and processing speed [the colour and form naming test from the "A quick test for cognitive speed" (AQT)] were associated with higher [18]F-AV-1451 retention ($P = 0.0093$ and $P = 0.037$), smaller hippocampal volume ($P = 0.032$ and $P = 0.048$) and thinner cortex ($P = 0.014$ and $P = 0.0050$). CSF T-tau or P-tau was not associated with cognitive measures (note that this was after adjusting for diagnosis).

**Table 1.  Study demographics.**

| | Controls | Prodromal AD | AD dementia |
|---|---|---|---|
| *N* | 30 | 14 | 39 |
| Age (year) | 74.7 (5.5) | 71.6 (6.3) | 71.3 (7.2) |
| Sex (F/M) | 15/15 | 10/4 | 18/21 |
| Education (year) | 11.3 (3.9) | 11.5 (3.8) | 11.9 (3.4) |
| MMSE | 29.3 (0.8) | 24.9 (2.6) | 21.1 (5.0) |
| ADAS-cog delayed recall | 2.2 (1.4) | 6.3 (2.4) | 8.4 (2.0) |
| AQT colour and form | 64 (13) | 89 (36) | 97 (38) |
| CSF Aβ42 (ng/l) | 682 (188) | 432 (83) | 393 (115) |
| Aβ (+/−) | 15/15 (50%) | 14/0 (100%) | 39/0 (100%) |
| CSF T-tau (ng/l) | 404 (154) | 675 (183) | 788 (325) |
| CSF P-tau (ng/l) | 52 (17) | 86 (17) | 96 (40) |

Continuous data shown as mean (standard deviation). MMSE, mini-mental state examination; ADAS-cog, Alzheimer's disease Assessment scale-cognitive subscale; AQT, A Quick Test of Cognitive Speed for Dementia; CSF, cerebrospinal fluid.

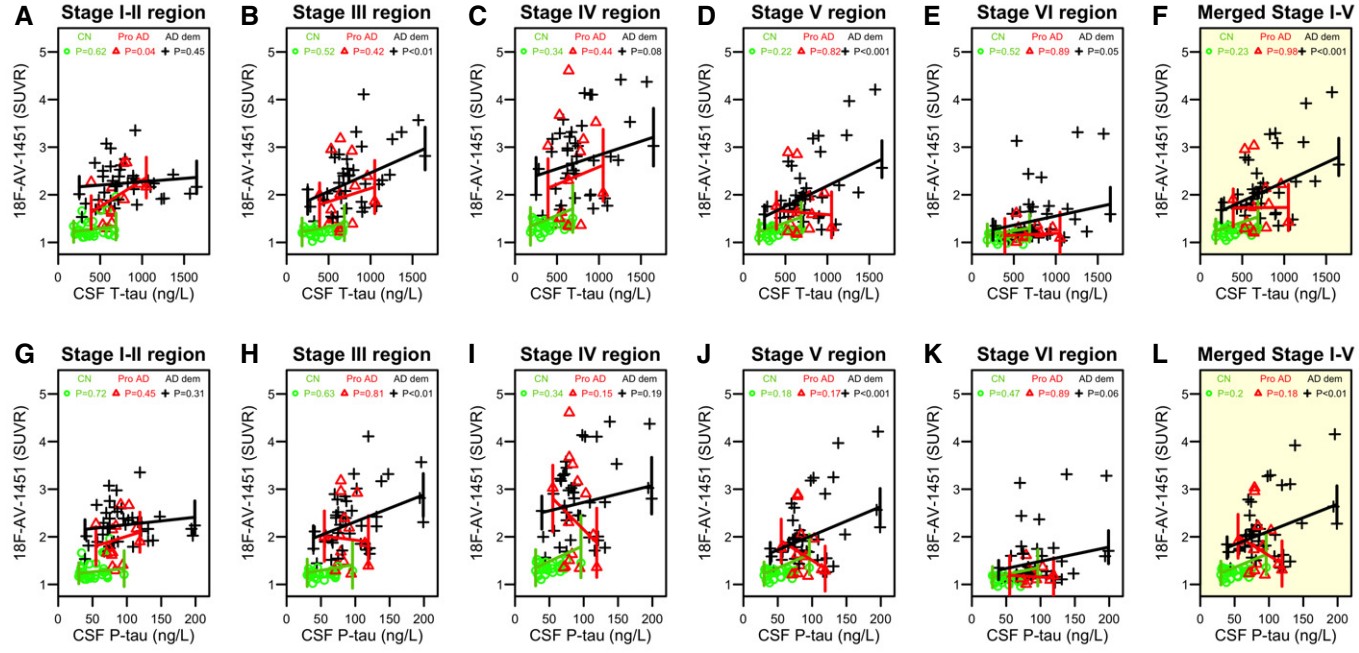

**Figure 1. Correlations between regional $^{18}$F-AV-1451 and CSF tau biomarkers.**

A–L  Observed data and correlations for CSF T-tau by regional $^{18}$F-AV-1451, and for CSF P-tau by regional $^{18}$F-AV-1451. The y-axes show $^{18}$F-AV-1451 data in the *a priori* defined tau stage I–II, III, IV, V and VI regions, and in the merged tau stage I–V region. Each symbol represents one person. The fitted lines are from linear regression models adjusted for the interaction between the CSF biomarker and diagnosis, and age, sex and time between lumbar puncture and PET scan. P-values from these models are shown, extracted for each diagnostic group (CN, controls; Pro AD, prodromal AD; AD dem, AD dementia). Corresponding β-coefficients are presented in Table EV1.

## CSF tau biomarkers are increased in preclinical AD

We next tested if $^{18}$F-AV-1451, CSF T-tau and P-tau varied by Aβ-status in controls. CSF Aβ42 < 650 ng/l was used as cut-off for Aβ-positivity (Palmqvist *et al*, 2014). There were 15 Aβ-positive and 15 Aβ-negative controls. Aβ-positive controls were classified as preclinical AD. CSF T-tau (β = 164.5, P = 0.0021) and P-tau (β = 12.6, P = 0.0043) were higher in preclinical AD (Fig 5A and B). There were also trends for higher $^{18}$F-AV-1451 retention in the tau stage III–V regions in preclinical AD, but the differences were not significant (Fig EV3A–F). All these models were adjusted for age and sex. The findings were the same when $^{18}$F-flutemetamol PET Aβ-imaging was used to define Aβ-positivity (Aβ-positivity was defined by a composite signal, as explained in Palmqvist *et al* (2014)). In this analysis, there were 16 Aβ-positive and 14 Aβ-negative controls [group differences were seen for CSF T-tau (P = 0.00034) and CSF P-tau (P = 0.022), but not for $^{18}$F-AV-1451 (P = 0.12–0.94)]. Since all prodromal AD and AD dementia patients were Aβ-positive, we could not test effects of Aβ in those groups.

## Sensitivity of tau biomarkers for Alzheimer's disease

All analyses described above were done using continuous tau measures. We also explored dichotomous tau measures, to test the sensitivity of tau biomarkers for AD. We used the mean value plus two standard deviations in the Aβ-negative controls to define

cut-offs for each biomarker. Sensitivity was calculated for preclinical AD, prodromal AD and AD dementia.

For $^{18}$F-AV-1451 in tau stage I–II, III, IV, V and VI regions, the cut-offs were 1.55, 1.35, 1.57, 1.40 and 1.30 SUVR, respectively [note that this was for partial volume error (PVE)-corrected data]. Figure 6A shows the sensitivity for preclinical AD, prodromal AD and AD dementia for $^{18}$F-AV-1451 positivity in different tau stage regions. In each region, the sensitivity increased from preclinical AD, to prodromal AD and AD dementia. For example, $^{18}$F-AV-1451 positivity in the tau stage III region had 13% sensitivity for preclinical AD, 86% sensitivity for prodromal AD and 100% sensitivity for AD dementia. $^{18}$F-AV-1451 had lower sensitivity in more advanced tau stage regions. In the tau stage VI region, $^{18}$F-AV-1451 positivity had 0% sensitivity for preclinical AD, 29% sensitivity for prodromal AD and 46% sensitivity for AD dementia.

Figure 6B shows sensitivities for the $^{18}$F-AV-1451 tau stage I–V composite region, CSF T-tau and P-tau. The tau stage I–V composite region (cut-off > 1.41 SUVR) had 0% sensitivity for preclinical AD, 64% sensitivity for prodromal AD and 92% sensitivity for AD dementia. CSF T-tau (cut-off > 542 ng/l) had 40% sensitivity for preclinical AD, 71% sensitivity for prodromal AD and 80% sensitivity for AD dementia. CSF P-tau (cut-off > 81 ng/l) had 0% sensitivity for preclinical AD, 50% sensitivity for prodromal AD and 54% sensitivity for AD dementia.

For external validation of the CSF cut-offs, we also calculated corresponding cut-offs for CSF T-tau and P-tau in an independent

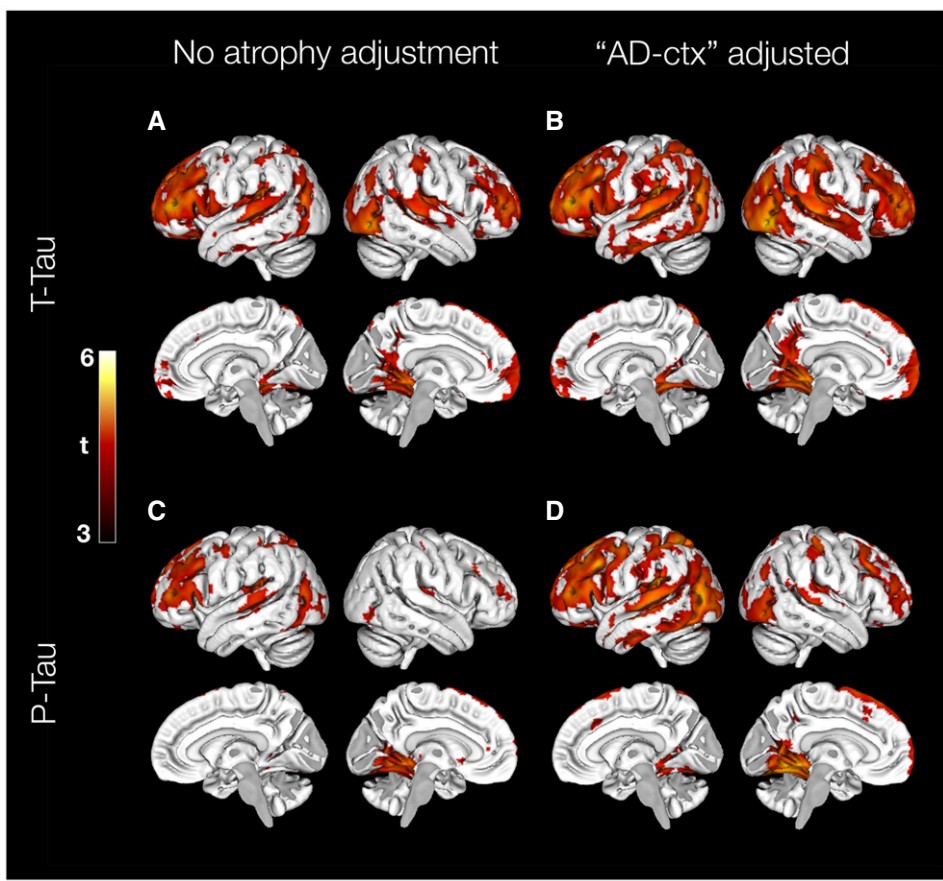

**Figure 2. Voxel-wise associations of $^{18}$F-AV-1451 and CSF tau biomarkers.**

A–D Associations of CSF T-tau (A) and P-tau (C) on $^{18}$F-AV-1451 uptake, adjusted for age, sex, diagnosis and time between lumbar puncture and PET scan. Associations of CSF T-tau (B) and P-tau (D) on $^{18}$F-AV-1451 uptake when also adjusting for atrophy (using cortical thickness in the "AD-cortex" region). All results are shown at a statistical significance threshold of $P < 0.001$ (uncorrected).

control population from the Swedish BioFINDER study (described in detail in Gustavsson *et al*, 2015), consisting of 246 Aβ-negative participants (Aβ-negativity defined using CSF Aβ42 > 650 ng/l). In this population, the CSF T-tau cut-off was 530 ng/l, and the CSF P-tau cut-off was 83 ng/l.

**Concordance between CSF and PET tau biomarkers differ by region and clinical stage**

Finally, we calculated the concordance between dichotomous CSF T-tau or P-tau and $^{18}$F-AV-1451 in the different tau stage regions (Fig 7A–H).

Most cases were classified concordantly by CSF tau biomarkers and $^{18}$F-AV-1451, but we noted some situations where discordance was common. For example, when comparing CSF T-tau with $^{18}$F-AV-1451 in the tau stage V region in the overall population (Fig 7A), most people were either negative (CSF⁻PET⁻ 31%) or positive (CSF⁺PET⁺ 45%) on both modalities, but isolated CSF T-tau positivity (CSF⁺PET⁻ 18%) was more common than isolated CSF T-tau negativity (CSF⁻PET⁺ 11%). This pattern was even more pronounced when comparing CSF T-tau with $^{18}$F-AV-1451 in the tau

stage VI region (CSF⁻PET⁻ 37%, CSF⁺PET⁻ 36%, CSF⁻PET⁺ 5%, CSF⁺PET⁺ 22%).

We also noted differences across the clinical stages of AD. In preclinical AD, several subjects had positive CSF T-tau and negative $^{18}$F-AV-1451 (Fig 7B). For example, when comparing CSF T-tau and $^{18}$F-AV-1451 in the tau stage I–II region, a majority were either negative for both modalities or isolated positive for CSF T-tau (CSF⁻PET⁻ 53%, CSF⁺PET⁻ 27%, CSF⁻PET⁺ 7%, CSF⁺PET⁺ 13%). In prodromal AD, several subjects were also isolated CSF tau positive (Fig 7C). For example, when comparing CSF T-tau and $^{18}$F-AV-1451 in the tau stage V region, 36% were isolated CSF T-tau positive, but only 14% were isolated CSF T-tau negative (CSF⁻PET⁻ 14%, CSF⁺PET⁻ 36%, CSF⁻PET⁺ 14%, CSF⁺PET⁺ 36%). In AD dementia, the concordance rates were generally higher than in the preclinical and prodromal stages of AD. But some subjects remained negative for CSF T-tau or P-tau despite having positive $^{18}$F-AV-1451 retention. For example, all AD dementia cases had positive $^{18}$F-AV-1451 in the tau stage III region, but 20% of them had negative CSF T-tau (Fig 7D) and 46% had negative CSF P-tau (Fig 7H).

The findings differed for T-tau and P-tau, since discordant cases for CSF T-tau were often $^{18}$F-AV-1451 negative (Fig 7A–D), while

 

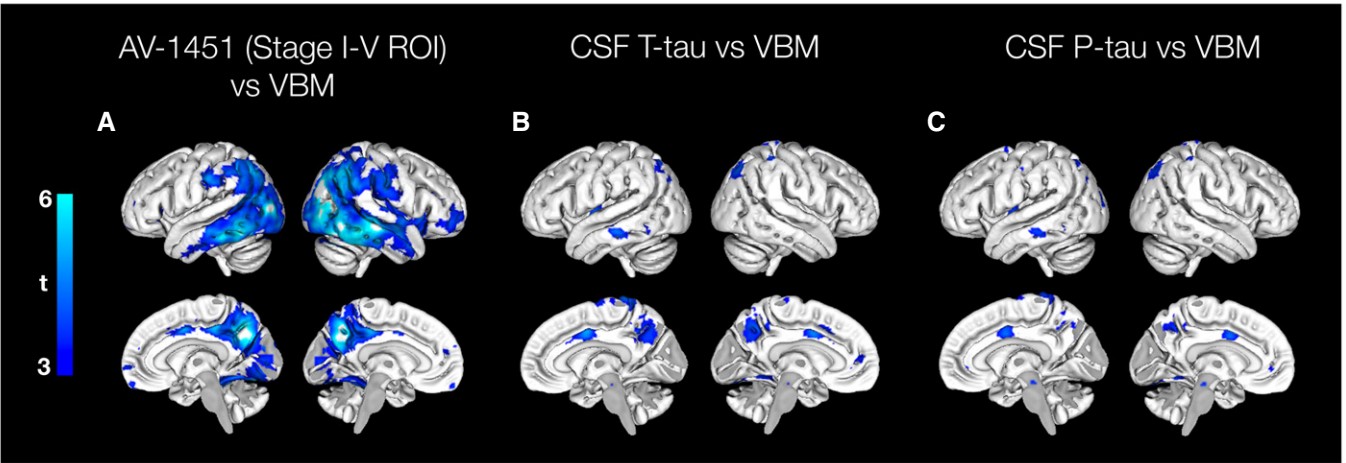

**Figure 3.  Associations between tau biomarkers and neurodegeneration.**

A    Voxel-wise association between GM intensity and [18]F-AV-1451 uptake in the tau stage I–V composite region, adjusted for age, sex and diagnosis. The model yielded suprathreshold clusters in the posterior lateral temporal lobes, lateral parietal and occipital cortex, as well as the precuneus at a statistical threshold of $P < 0.001$ (uncorrected).

B, C  Voxel-wise association between GM intensity and CSF T-tau or P-tau, adjusted for age, sex and diagnosis. The model with T-tau as the independent variable produced small, scattered significant clusters in the precuneus, the middle cingulate gyrus, parietal and left temporal cortex. The model with P-tau produced a similar pattern, except for the association in precuneus.

discordant cases for CSF P-tau were often [18]F-AV-1451 positive (Fig 7E–H).

## Discussion

This study of CSF T-tau, P-tau and [18]F-AV-1451 in AD resulted in several novel findings. First, CSF T-tau and P-tau correlated moderately with [18]F-AV-1451 in AD dementia, but not in controls and only rarely in prodromal AD. Second, [18]F-AV-1451 was closely associated with neurodegeneration and cognitive decline, while CSF tau biomarkers were only mildly associated with neurodegeneration, and not with cognition (when adjusted for diagnosis and other confounders). Third, CSF T-tau and P-tau, but not [18]F-AV-1451, were increased in preclinical AD. However, increased CSF T-tau and P-tau levels were not ubiquitous, since even a proportion of AD dementia patients had normal CSF tau levels, even though [18]F-AV-1451 retention was virtually always increased in the dementia stage. Fourth, the concordance between dichotomous CSF tau and [18]F-AV-1451 measures was greatest in the AD dementia group. Together, this supports a model where CSF T-tau and P-tau primarily behave as disease state markers in AD, indicating the presence of the AD process, while [18]F-AV-1451 primarily behave as a disease stage marker, with continuous build-up throughout the clinical stages of the disease. Note that we adhere to the definitions of disease state and stage biomarkers as presented in (Blennow & Hampel, 2003). With this terminology, disease state biomarkers reflect the intensity of a disease process, while disease stage biomarkers reflect how far the process has progressed. The early plateauing of CSF T-tau and P-tau (Andreasen et al, 1999; Blennow et al, 2007; Zetterberg et al, 2007; Rosén et al, 2012) may make these measures suitable as disease state biomarkers. In contrast, the continuous increase in [18]F-AV-1451 retention throughout the disease (Brier et al, 2016;

Chhatwal et al, 2016; Gordon et al, 2016), when also atrophy and cognitive symptoms continue to develop, may make it suitable as a disease stage biomarker.

The finding that CSF T-tau and P-tau may be increased without increased [18]F-AV-1451 PET retention suggests that CSF T-tau, CSF P-tau and [18]F-AV-1451 partly reflect different processes. Specifically, the increased CSF T-tau and P-tau levels in preclinical AD, in the absence of increased [18]F-AV-1451 retention, are compatible with an increased release of tau from neurons in early stages of AD, before overt tau pathology has accumulated. This may influence the operationalization of tau biomarkers in AD, depending on whether the aim is to identify very early cases or having high specificity in later disease stages. One intriguing hypothesis is that a fraction of CSF tau may partly reflect active secretion from neurons (Maia et al, 2013). We noted that CSF T-tau and P-tau were *positively* associated with cortical thickness when models were adjusted for [18]F-AV-1451, which may reflect a normal release of tau to CSF that is easier to detect when the models are adjusted for the presence of tau pathology. It is logical that a fraction of CSF tau reflects normal neuronal metabolism, since some CSF tau is ubiquitously present in all humans, irrespective of disease.

Another important finding was that some AD dementia patients had relatively low CSF tau levels. Since these patients were Aβ-positive and had increased [18]F-AV-1451 compared to controls, we do not believe that they were misdiagnosed. Instead, the finding highlights that there is a variability in CSF tau release among AD patients. This may be related to the rate of clinical progression of the disease (Wallin et al, 2010; Degerman Gunnarsson et al, 2016). We previously found that CSF tau measures were higher in MCI patients who converted to AD dementia within a few years, compared to patients who converted to AD dementia only after 5–10 years (Buchhave et al, 2012). Taken together, these studies suggest that some patients may have a more aggressive form of the

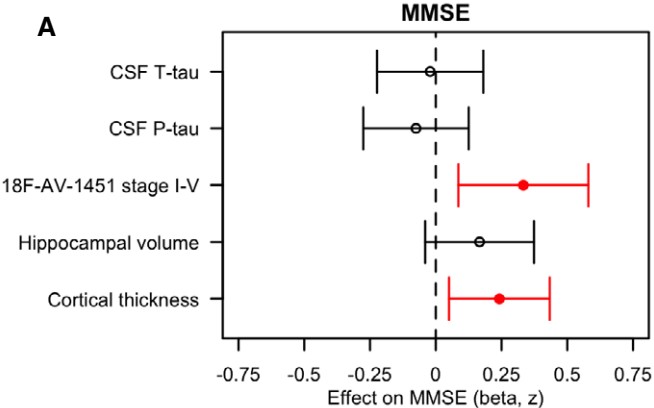

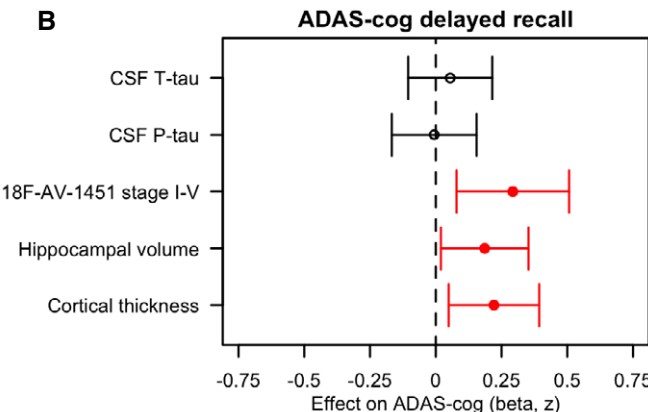

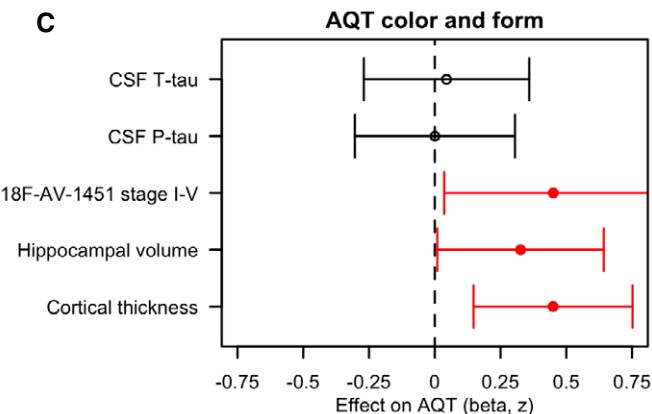

**Figure 4. Associations between tau biomarkers and cognition.**

A–C Associations of CSF T-tau, P-tau, [18]F-AV1451 retention in the stage I–V composite region, hippocampal volume and cortical thickness (in the temporal lobe, "AD-cortex", see Materials and Methods for details) with the cognitive tests MMSE, ADAS-cog delayed recall, and AQT colour and form naming. The estimates are β-coefficients with 95% confidence intervals (error bars) from linear regression models adjusted for age, sex, diagnosis, education and (for hippocampal volume) intracranial volume. All data were scaled and centred. If necessary, the signs were inverted to make a more pathological biomarker associated with greater cognitive impairment. Significant associations are shown in red. In sum, higher [18]F-AV1451 retention and thinner cortex were associated with worse MMSE, ADAS-cog delayed recall and AQT; and smaller hippocampal volume was associated with worse ADAS-cog delayed recall and AQT.

disease, which is characterized both by higher CSF tau levels and a more rapidly progressive clinical course.

Neurodegeneration and cognition were strongly correlated with [18]F-AV-1451, but not with CSF T-tau and P-tau. Note that all these tests were done in the pooled cohort, but adjusted for diagnosis. When not adjusting for diagnosis, both CSF T-tau and P-tau were strongly correlated with neurodegeneration and cognitive impairment, as expected (data not shown). We consider diagnosis of AD to be a confounding factor for the associations between tau biomarkers, neurodegeneration and cognition, and by adjusting for diagnosis, we believe that we may provide more robust estimates of the correlations. Measures that are mainly associated with disease state, such as CSF T-tau and P-tau, loose their correlations with neurodegeneration and cognition when diagnosis is considered. In contrast, measures that are also associated with disease stage, such as [18]F-AV-1451, retain a correlation with neurodegeneration and cognition even when diagnosis is considered.

The overall concordance rates between CSF tau biomarkers and [18]F-AV-1451 were around 70%. This is ~10–15% lower than the concordance for amyloid CSF and PET biomarkers (Palmqvist *et al*, 2014; Mattsson *et al*, 2015; Janelidze *et al*, 2016). One possible explanation for the discrepancy between CSF tau biomarkers and [18]F-AV-1451, in comparison with CSF Aβ and Aβ PET, is that Aβ biomarkers cluster into more well-characterized bimodal distributions compared to tau biomarkers. This suggests that rates of Aβ may accelerate/decelerate more than tau at a specific boundary. The issues of concordance between CSF tau and [18]F-AV-1451 may have implications for criteria that combines tau biomarkers with biomarkers of Aβ-pathology and neurodegeneration (Dubois *et al*, 2014; Jack *et al*, 2016a). Hypothetically, CSF tau biomarkers may provide unique information in early stages of the disease, but [18]F-AV-1451 appears more robust at the dementia stage.

One of the main conclusions of this study is that CSF T-tau and P-tau primarily behave as disease state markers for AD. However, although CSF T-tau and P-tau were highly correlated, they may not mark exactly the same pathological state, since CSF T-tau is elevated in AD but also in other diseases, including stroke and Creutzfeldt-Jakob's disease, while CSF P-tau is more specific to AD (Mattsson, 2011). CSF P-tau has been suggested to be more closely related to brain tau pathology than CSF T-tau (Blennow *et al*, 2010), since histopathology studies have primarily found correlations for P-tau with tangle load (Buerger *et al*, 2006; Seppälä *et al*, 2012) [although one study found correlations also for T-tau (Tapiola *et al*, 2009) and one did not find correlations between tau pathology and CSF tau (Engelborghs *et al*, 2007)]. We did not find evidence for a stronger relationship with [18]F-AV-1451 retention for CSF P-tau than for T-tau. On the contrary, the associations were slightly more pronounced for T-tau than for P-tau. It is possible that CSF P-tau would have been more closely related to [18]F-AV-1451 if we also included other dementias, since the relationship between [18]F-AV-1451, CSF T-tau and P-tau may differ between tauopathies.

One limitation of our study is the lack of neuropathological confirmation of tau pathology. We tried to overcome this by defining tau positivity in Aβ-negative controls, who were very unlikely to have substantial tau pathology. For the [18]F-AV-1451 regional analyses, we used a staging system that is similar to the Braak staging system, but we acknowledge that tau PET to stage AD pathology is

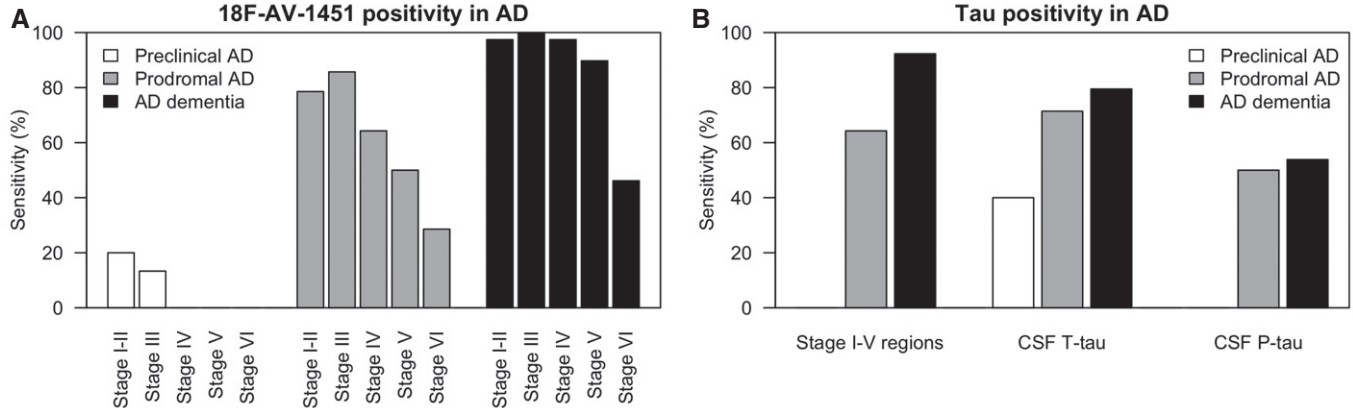

**Figure 5.  Sensitivity of tau biomarkers for Alzheimer's disease.**

A, B   The sensitivity of [18]F-AV-1451 in different tau stage regions for preclinical AD, prodromal AD and AD dementia, and the sensitivity of [18]F-AV-1451 in the tau stage I–V composite region, CSF T-tau and P-tau. Cut-offs were defined using the mean levels plus two standard deviations in Aβ-negative controls, as explained in the main text.

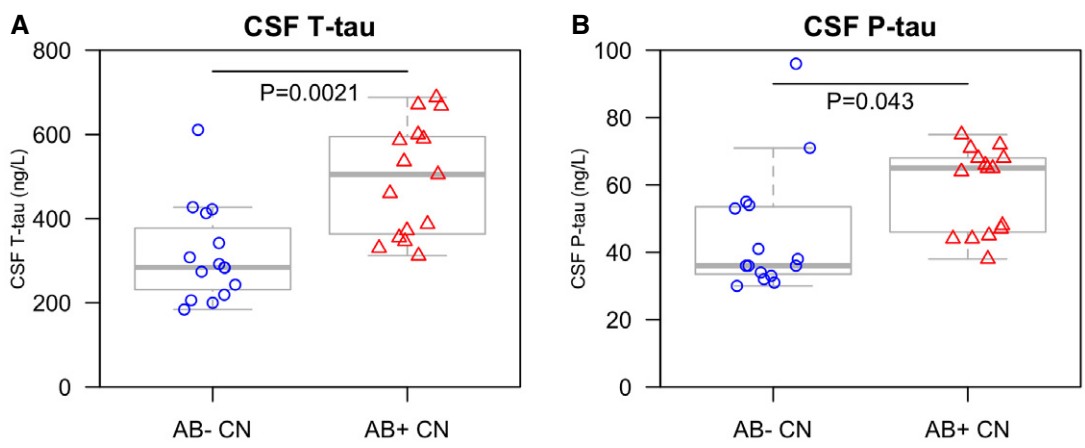

**Figure 6.  CSF tau biomarkers in preclinical AD.**

A, B   CSF T-tau and P-tau were compared between Aβ-negative and Aβ-positive controls. Aβ-positivity (i.e. preclinical AD) was associated with higher levels of CSF T-tau and P-tau, adjusted for age and sex. We also compared CSF T-tau and P-tau between Aβ-negative and Aβ-positive subjects when including all subjects with a negative [18]F-AV-1451 signal in the tau stage I–V tau composite region (30 CN, five prodromal AD and three AD dementia patients). The Aβ-dependent effects for CSF T-tau and P-tau were even greater in this group, with higher CSF T-tau (β = 222, t = 3.9, P = 0.00048) and P-tau (β = 22.7, t = 2.9, P = 0.0054) associated with Aβ-positivity. The boxplots show medians and interquartile ranges.

still developing and other staging systems may produce different results (Jack *et al*, 2016b). We used the Geometric Transfer Method (GTM) method for PVE correction, which results in 10-15% higher SUVR values compared to other methods (Schöll *et al*, 2016). With non-PVE-corrected data, the [18]F-AV-1451 cut-offs for positive signals were 1.28, 1.23, 1.31, 1.16, 1.09 and 1.18 in the tau stage regions I–II, III, IV, V, VI, and the I–V composite, respectively, which were similar to previously proposed cut-offs (Jack *et al*, 2016b; Ossenkoppele *et al*, 2016). Another limitation is that the sample size for some subgroups was small. This may have reduced the power to detect subtle increases in [18]F-AV-1451 in preclinical AD. A much larger study (*N* = 490) recently reported increased [18]F-AV-1451 uptake in Aβ-positive controls (Vemuri *et al*, 2016). However, there may also be cohort-dependent differences, since a previous smaller study also

reported increased [18]F-AV-1451 uptake in Aβ-positive controls (Schöll *et al*, 2016). Finally, we acknowledge that [18]F-AV-1451 is not completely specific for tau, although post-mortem studies have showed that it has high affinity for AD-type tau lesions (Smith *et al*, 2016a), and not to other tau lesions (seen in other tauopathies) or other protein inclusions (Marquié *et al*, 2015). Off-target binding may occur, for example, in the basal ganglia (Smith *et al*, 2017) and in the choroid plexus, which is close to the hippocampal formation. This was a reason for us to exclude the hippocampus from the [18]F-AV-1451 analyses.

In conclusion, we found that CSF T-tau and P-tau correlated with [18]F-AV-1451 in AD dementia, but [18]F-AV-1451 was more strongly correlated with neurodegeneration and cognitive decline than the CSF tau biomarkers were. Different dynamics of tau biomarkers in

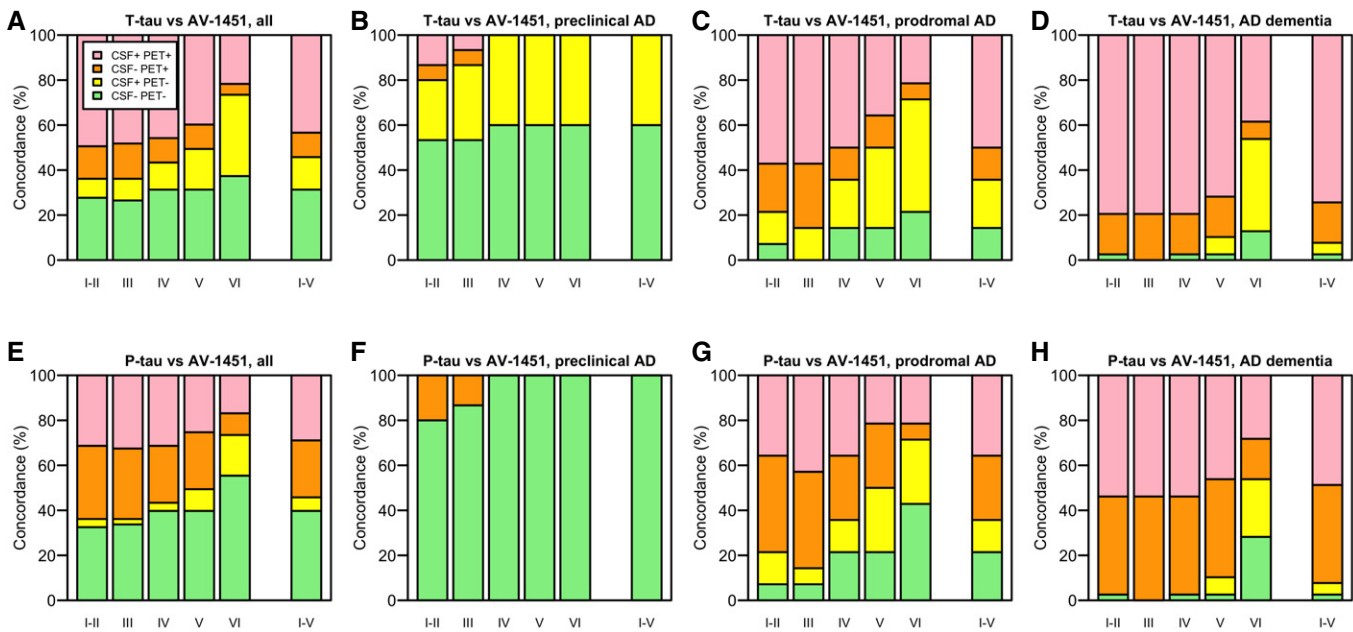

**Figure 7. Concordance between CSF and PET tau biomarkers.**

A–H   Concordance between CSF T-tau and [18]F-AV-1451, and CSF P-tau and [18]F-AV-1451. Panels (A) and (E) include all subjects, and panels (B–D) and (F–H) include AD patients at different disease stages. The cut-off for positive CSF T-tau was > 542 ng/l. The cut-off for positive CSF P-tau was > 81 ng/l. For [18]F-AV-1451, the cut-offs were > 1.55, 1.35, 1.57, 1.40 and 1.30 SUVR, for tau stage I–II, III, IV, V and VI regions, respectively, and > 1.41 SUVR for the tau stage I–V composite region. Note that every participant is represented in each category on the *x*-axes, since every participant was classified as positive or negative for [18]F-AV-1451 in every tau stage region.

different stages of AD may have implications for their use when stratifying AD patients. Future studies on longitudinal tau measures will be important to understand how they develop in AD, and to detect effects of disease-modifying treatments.

# Materials and Methods

### Subjects

The study population stemmed from three cohorts from the prospective and longitudinal Swedish BioFINDER study (www.biofinder.se). We included 30 cognitively normal elderly participants, who were eligible for inclusion if they: (i) were aged ≥ 60 years old, (ii) scored 28–30 points on the MMSE at the screening visit, (iii) did not fulfil the criteria of MCI or any dementia, and (iv) were fluent in Swedish. The exclusion criteria were the following: (i) presence of significant neurologic or psychiatric disease (e.g. stroke, Parkinson's disease, multiple sclerosis, major depression), (ii) significant systemic illness making it difficult to participate, (iii) refusing lumbar puncture, or (iv) significant alcohol abuse. In the second cohort, 14 patients with MCI due to AD were enrolled at the Memory Clinic of the Skåne University Hospital, Sweden. The inclusion criteria were the following: (i) referred to the memory clinics because of cognitive impairment, (ii) not fulfilling the criteria for dementia, (iii) an MMSE score of 24–30 points, (iv) objective memory impairment according to delayed word list recall, (v) age 60–80 years, (vi) abnormal CSF Aβ42 levels, and (vii) fluent in Swedish. The exclusion criteria were the following: (i) cognitive impairment explained by another

condition (other than prodromal dementia), (ii) significant systemic illness making it difficult to participate, (iii) refusing lumbar puncture, or (iv) significant alcohol abuse. In the last cohort, we included 39 patients with AD dementia at baseline, who were recruited at the Memory Clinic, Skåne University Hospital. The patients were assessed by a medical doctor specialized in dementia disorders. All cases met the DSM-IIIR criteria for dementia (American Psychiatric Association, & American Psychiatric Association, Work Group to Revise DSM-III, 1987) as well as the NINCDS-ADRDA criteria for AD (McKhann *et al*, 1984). The exclusion criteria were the following: (i) significant systemic illness making it difficult to participate, and (ii) significant alcohol abuse. The diagnosis of MCI due to AD and AD dementia were established by physicians who were blinded to the [18]F-AV-1451 PET and CSF T-tau and P-tau data (S.P. and O.H.). All available subjects from the BioFINDER study who fulfilled criteria were included.

### Cognitive measures

We used the MMSE as a measure of general cognition, the delayed recall memory test from ADAS-cog (list learning, 10 items) as a measure of memory, and the colour and form naming test from the AQT as a test of attention and processing speed (Wiig *et al*, 2010).

### CSF biomarkers

CSF samples were derived by lumbar puncture (Palmqvist *et al*, 2014). Samples were analysed at the Clinical Neurochemistry

Laboratory in Mölndal, Sweden, for T-tau, P-tau and Aβ42 using commercially available ELISAs (INNOTEST, Fujirebio, Ghent, Belgium). All CSF samples were analysed using clinical practice procedures, with analyses performed by board-certified technicians blinded to clinical data, following detailed procedures to assure analytical precision and long-term stability, including batch bridging between old and new batches of ELISA plates, general laboratory procedures (e.g. calibration of pipettes and preventive service of instruments) and strict criteria for approval of calibration curves and internal QC samples, following the Westward multi rules, as described previously in detail (Palmqvist *et al*, 2014). Long-term stability of the results was monitored in the Alzheimer's Association QC program (Mattsson *et al*, 2013). The approval limits for the two internal QC CSF samples run at two positions on each plate were 12.0% for Aβ42, 9.3% for T-tau and 9.8% for P-tau for the normal QC sample, and 11.0% for Aβ42, 10.0% for T-tau and 9.8% for P-tau for the AD-like QC sample.

### MRI imaging and processing

T1-weighted imaging was performed on a 3T MR scanner (Siemens Tim Trio 3T, Siemens Medical Solutions, Erlangen, Germany), producing a high-resolution anatomical MP-RAGE image (TR = 1,950 ms TE = 3.4 ms, 1 mm isotropic voxels and 178 slices) for further use in volumetric analysis, template normalization and coregistrations. The anatomical scan was normalized to MNI152 space (Grabner *et al*, 2006) with a diffeomorphic transform and the Advanced Normalization Tools (ANTs) toolbox (Avants *et al*, 2014) for further use in the PET processing pipeline (see below; ANTs was used for all coregistrations). Cortical reconstruction and volumetric segmentation were performed with the FreeSurfer image analysis pipeline v5.3 (http://surfer.nmr.mgh.harvard.edu/). Briefly, the T1-weighted images underwent correction for intensity homogeneity (Sled *et al*, 1998), removal of non-brain tissue (Ségonne *et al*, 2004), and segmentation into grey matter (GM) and white matter (WM) with intensity gradient and connectivity among voxels (Dale *et al*, 1999; Fischl & Dale, 2000; Fischl *et al*, 2002, 2004b). Cortical modelling allowed parcellation of the cerebral cortex into units with respect to gyral and sulcal structure (Fischl *et al*, 2004a; Desikan *et al*, 2006). Cortical thickness was measured as the distance from the GM/WM boundary to the corresponding pial surface (Fischl & Dale, 2000). Reconstructed data sets were visually inspected for accuracy, and segmentation errors were corrected.

FreeSurfer processed data was used to calculate average hippocampal volumes and a composite cortical thickness measure of temporal regions vulnerable to atrophy in AD ["AD-cortex", which included surface-weighted thickness measures of bilateral entorhinal cortex, inferior temporal cortex, middle temporal cortex and fusiform cortex (based on Jack *et al* (2015))].

VBM as implemented in SPM12 (http://www.fil.ion.ucl.ac.uk/spm) was used to evaluate GM intensity differences as a measure of GM atrophy. In preparation, all individual T1-weighted MR images were segmented into tissue classes, and the GM segmentations subsequently warped into a common MNI152 standard space (using a cohort-specific template created with the DARTEL toolbox and Jacobian scaling to estimate GM intensity). The resulting maps were smoothed with an 8 mm full width at half maximum (FWHM) Gaussian kernel. Intracranial volume was calculated by summating the

volumes of GM, WM and CSF space based on the SPM tissue segmentations.

### Tau PET imaging and processing

[18]F-AV-1451 was synthesized at Skåne University Hospital, Lund, as described previously (Hahn *et al*, 2016). PET scans were performed on a GE Discovery 690 PET scanner (General Electric Medical Systems) as dynamic scans using LIST-mode 80–120 min after a bolus injection of 370 MBq [18]F-AV-1451. Low-dose CT scans for attenuation correction were performed immediately prior to the PET scans. PET data were reconstructed into 5-min frames using an iterative Vue Point HD algorithm with six subsets, 18 iterations with 3-mm filter and no time-of-flight correction. The dynamic scans were motion-corrected using AFNI's 3dvolreg (Cox, 1996), time-averaged and rigidly coregistered to the skull-stripped MRI scan. PVE correction was performed using the GTM as described in (Rousset *et al*, 1998) using the FreeSurfer parcellations, smoothed with 5-mm FWHM to calculate transfers across regions of interest (ROI) borders. Furthermore, we combined GTM method with region-based voxel-wise (RBV; Thomas *et al*, 2011) method, producing voxel-wise PVE corrections, which were warped to standard MNI152 space using the MRI normalization for further voxel-wise analysis.

The FreeSurfer parcellation in the MR space of the anatomical scan was then applied to the processed, coregistered and time-averaged PET image to extract regional uptake values. We created AV-1451 standardized uptake value (SUV) images based on mean uptake over 80–120 min postinjection and normalized to uptake in a GM-masked cerebellum reference region to create voxel-wise SUVR images in each participant's MRI native space (Smith *et al*, 2017).

Due to the addition of [18]F-AV-1451 PET imaging well after the start of the Swedish BioFINDER study, CSF was obtained on average 6.5 (mean, standard deviation 8.9) months, 18.6 (16.0) months and 18.7 (18.2) months, respectively, for controls, prodromal AD, and AD dementia prior to [18]F-AV-1451 imaging. This delay was examined as a potential covariate in statistical models.

### Regional PET analyses

One of our main aims was to test correlations between CSF tau biomarkers and regional [18]F-AV-1451 uptake. To avoid a multiple comparison problem, we restricted the analyses to *a priori* defined ROIs. We used a previously described protocol to aggregate Free-Surfer ROIs by their involvement in different stages of tau spread, as determined using [18]F-AV-1451 PET in cases with different stages of AD (Cho *et al*, 2016), and these stages roughly corresponds to the Braak staging approach (Braak & Braak, 1991). This resulted in a set of non-overlapping ROIs, labelled the tau stage I–II region, the tau stage III region, the tau stage IV region, the tau stage V region and the tau stage VI region. The included ROIs in each tau stage region are listed in the results section. To capture overall tau pathology, we did a composite of tau stage I–V regions (we excluded the tau stage VI region since [18]F-AV-1451 was generally were low in this region, even among AD dementia patients). For each tau stage region, the signal was calculated as the sum of the volume-adjusted [18]F-AV-1451 signal in each included ROI. Note that with this

terminology, each "tau stage" refers to a specific part of the brain. The system was not used to assign a single tau stage to each subject. Instead, each subject had measures of [18]F-AV-1451 in each of the tau stage regions.

In some analyses, we used a dichotomous assignment of positive or negative [18]F-AV-1451, with cut-offs defined as the mean signal plus two standard deviations in Aβ-negative controls, which is in agreement with how cut-offs for Aβ PET have been established (Clark *et al*, 2011).

## Statistical analyses

Correlations between CSF T-tau and P-tau were tested by Spearman's correlation, and linear regression models adjusted for age, sex and diagnosis.

Linear regression was used to test associations between regional [18]F-AV-1451 (as outcome) and CSF T-tau or P-tau (as predictors), adjusted for the interaction between CSF tau and diagnosis, age, sex and time lag between lumbar puncture and PET scan. Note that the analyses were done on the pooled group. For the results presented in Fig 1, we used the interaction between biomarker and diagnosis to extract the correlations within each diagnostic group.

We also used linear regression models to test associations between CSF T-tau, P-tau, regional [18]F-AV-1451 and imaging measures of neurodegeneration and cognitive outcomes, adjusted for diagnosis, age, sex, education and (for hippocampal volume) intracranial volume. For cognitive outcomes, all biomarker and imaging measures were standardized to z-scores, and inverted if necessary, to facilitate comparisons across modalities and cognitive outcomes.

The model assumptions were assessed by evaluating normality and homoscedasticity of residuals with q–q plots and plots of residuals versus fitted values. All tests were two-sided. Significance was determined at $P < 0.05$. All statistics described above were done using R (v. 3.2.3, The R Foundation for Statistical Computing).

Associations between CSF biomarkers and [18]F-AV-1451 were further analysed using voxel-wise multiple regression models as implemented in SPM12. All images had been transformed into common MNI152 space by using transformation measures from warping the coregistered MRI scans to the 2 mm FSL MNI152 MRI template (http://www.fmrib.ox.ac.uk/fsl) and smoothed with an 8-mm FWHM kernel prior to analysis. Finally, we tested voxel-wise associations between GM intensity using VBM measures described above and measures of CSF tau and the [18]F-AV-1451 signal in the tau stage I–V composite region. All resulting *t*-maps were thresholded to present results with a $P < 0.001$ height, uncorrected for family-wise error.

## Ethics

All participants gave written informed consent to participate in the study. Ethical approval was given by the Ethical Committee of Lund University, Lund, Sweden, and all the methods were carried out in accordance with the approved guidelines. [18]F-AV-1451 PET imaging approval was obtained from the Swedish Medicines and Products Agency and the local Radiation Safety Committee at Skåne University Hospital, Sweden.

**The paper explained**

**Problem**

Tau pathology is a key feature of Alzheimer's disease, but the relationships between cerebrospinal fluid tau and tau PET measures are unclear.

**Results**

We found that cerebrospinal fluid tau biomarkers and tau PET using the tracer [18]F-AV-1451 were moderately correlated, but also that they had different relationships to other features of Alzheimer's disease. Cerebrospinal fluid tau was increased already in preclinical Alzheimer's disease, but [18]F-AV-1451 was correlated to more advanced neurodegeneration and cognitive decline.

**Impact**

The findings suggest that cerebrospinal fluid tau and [18]F-AV-1451 are differently affected during different stages of Alzheimer's disease. This may have important consequences for the use of different types of tau biomarkers in clinical practice and in clinical trials of new disease-modifying treatments.

**Expanded View** for this article is available online.

## Acknowledgements

Work in the authors' laboratory was supported by the European Research Council, the Swedish Research Council, the Strategic Research Area MultiPark (Multidisciplinary Research in Parkinson's disease) at Lund University, the Swedish Brain Foundation, the Skåne University Hospital Foundation, the Swedish Alzheimer Foundation, the Marianne and Marcus Wallenberg Foundation, the Swedish Federal Government under the ALF Agreement, the Greta and Johan Kock Foundation for Medical Research, the Thelma Zoega Foundation for Medical Research, the Bundy Academy, the Emil and Maria Palm's Foundation and the Magnus Bergwall Foundation. The funding sources had no role in the design and conduct of the study, in the collection, analysis, interpretation of the data or in the preparation, review or approval of the manuscript. The precursor of AV-1451 was generously provided by Avid Radiopharmaceuticals.

## Author contributions

NM and OH designed the study, performed the main analyses and interpretation of data and drafted the manuscript. RS, SP, DH, TO, HZ, JJ and KB participated in the acquisition of the data. PSI, MS and OS participated in the analysis and interpretation of the data. MS; OS, RS, SP, PSI, DH, TO, HZ, JJ and KB revised the work critically for important intellectual content. All authors approved the final version to be published.

## Conflict of interest

Drs Mattsson, Schöll, Palmqvist, Insel, Strandberg, Hägerström, Ohlsson, Jögi and Smith report no conflicts of interest. Dr Hansson has served at advisory boards Eli Lilly, Roche Diagnostics and Fujirebio and received research support from GE Healthcare and Hoffmann La-Roche. Dr Blennow has served at advisory boards or as a consultant for Alzheon, Eli Lilly, Fujirebio Europe, IBL International, Novartis and Roche Diagnostics. Dr Zetterberg has served at advisory boards for Roche Diagnostics, Pharmasum Therapeutics and Eli Lilly. Drs Zetterberg and Blennow are co-founders of Brain Biomarker Solutions in Gothenburg AB, a GU Venture-based platform company at the University of Gothenburg.

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
