## [Review Process File · EMBO Molecular Medicine]

^{18}F -AV-1451 and CSF T-tau and P-tau as biomarkers in Alzheimer's disease

Niklas Mattson, Michael Schöll, Olof Strandberg, Ruben Smith, Sebastian Palmqvist, Philip S Insel, Douglas Hågerström, Tomas Ohlsson, Henrik Zetterberg, Jonas Jögi, Kaj Blennow, Oskar Hansson

Corresponding author: Niklas Mattsson, Lund University

Review timeline:	Submission date:	16 March 2017
	Editorial Decision:	24 April 2017
	Revision received:	28 April 2017
	Editorial Decision:	31 May 2017
	Revision received:	14 June 2017
	Accepted:	20 June 2017

Editor: Céline Carret

Transaction Report:

1st Editorial Decision

24 April 2017

Thank you for the submission of your manuscript to EMBO Molecular Medicine. We have now heard back from the three referees whom we asked to evaluate your manuscript. Although the referees find the study to be of potential interest, they also raise a number of concerns that need to be addressed in the next version of your article.

You will see from the comments pasted below, that all referees find the study potentially interesting while some design and analytic flaws are highlighted: correlations should be rethought, biomarkers vs. state vs stage markers terminology and rationale are not fully exploited, nor clearly defined; refs 2 and 3 picked up an odd patient classification that must be fixed; fig6 (and 7) are not clear enough for refs 2 and 3. Overall, while the study should be very valuable to the community, as presented it doesn't deliver because of insufficient data analyses and poor presentation.

We would welcome the submission of a revised version for further consideration and would like to encourage you to address all the criticisms raised as suggested to improve conclusiveness and clarity. Please note that EMBO Molecular Medicine strongly supports a single round of revision and that, as acceptance or rejection of the manuscript will depend on another round of review, your responses should be as complete as possible.

EMBO Molecular Medicine has a "scooping protection" policy, whereby similar findings that are published by others during review or revision (typically 3 months-revision) are not a criterion for rejection. Should you decide to submit a revised version, I do ask that you get in touch after three months if you have not completed it, to update us on the status.

I look forward to receiving your revised manuscript.

***** Reviewer's comments *****

Referee #1 (Comments on Novelty/Model System):

The authors have designed their study with an a priori knowledge on Braak stages and CSF Tau biomarkers. They should justify this approach since there is no evidence that Tau PET ligand or CSF Tau biomarkers reflect neurofibrillary degeneration. Correlation/Covariance of raw data may have also been of interest without any a priori approach.

Referee #1 (Remarks):

First, I must apologize for the delay in my review.

Mattsson and coll. have investigated the association between CSF Tau biomarkers and the Tau PET ligand AV1451 using linear regression models. They also analyzed associations between Tau markers (CSF and PET) and atrophy (VBM approaches) and between Tau markers and cognition (MMSE, ADAS-Cog).

This is an original study, well written and well done. Nevertheless, it needs some clarifications I have only minor comments:

- 1) First, it will be interesting to see any association without any a priori approach (w/o Braak stages, w/o cut-offs...) as in Suppl Fig1. The sample population is low and such stratification may lower statistical power but the choice of the authors has to be justified.
- 2) The positive PET signal seen in Fig 3 and Suppl. in the locus coeruleus is not included in any analyses. Since it was reported that Tau pathology starts in this area, the authors may include a Braak stage 0 to confirm (or not) this information.
- 3) The discussion needs to be more balanced: such Tau PET ligands are not fully specific and other targets have been identified.
- 4) In AD dementia, there is a subset of patients with high Tau concentrations (as suggested by SD in Table 1). Do they have a particular Tau PET distribution?
- 5) In figure 1, is each spot an individual person, a combination of the three CSF quantification? Why the merged graph does not show more values?

Referee #2 (Remarks):

The manuscript is extremely hard to read and therefore also difficult to review properly. After spending several hours for reading it twice, I still don't fully comprehend figures 6 and 7. I think, in the current presentation the manuscript is only readable for the group of authors and their immediate competitors. The subsections headings, figure title and even the manuscript title give absolutely no clue about the findings. There is not even a single verb in any of those headings. The result section is particularly difficult to read. The subsections don't start with a rationale and don't finish with a conclusion. In the legends the individual panels should be mentioned and described specifically. Nevertheless, the manuscript contains some interesting data on the time course of biomarkers in AD, which are largely hidden due to poor presentation.

The authors seem to misuse the word "disease state markers", which normally refers to a biomarker that is positive only in a certain disease stage. From the data it seems that both CSF-tau and PET-tau are disease state markers that become positive in different disease stages. If at all, the common terminology trait vs. state marker should be used.

The abbreviation "AD-ctx" is also confusing - why not call it atrophy? Does "T-tau ($\beta=0.00024$ [unit: ng/L], $p=0.015$) and P-tau ($\beta=0.0025$ [unit: ng/L], $p=0.0021$) had positive associations with AD-cortex in these adjusted models" mean more Tau is associated with thicker or thinner cortex? In the discussion the answer is finally revealed.

Tau staging should be explained in the result section or maybe even indicated in the figure 1 or 2. Do the results from Fig. 1 match with the voxel-based result from Fig 2?

Does stage IV mean patients with Stage IV PET pattern or does it denote the region that defines stage IV patients. I assume the latter is correct and "region IV" would be a less misleading term.

"Tau biomarkers and preclinical AD": The authors should define "Abeta status" in result section. In that context it could mean PiB positive or CSF-marker positive. In fact, including amyloid PET in the correlation analysis would be nice.

"Sensitivity of tau biomarkers for AD"

Are the figures mislabeled? The text mentions stages and dichotomous tau measures, but the figure shows continuous tau measures and nothing related to 18F-AV-1451 stages. How were the cut-offs determined? I didn't understand that part at all.

Page 8: "isolated CSF tau positivity was most common for stage VI tau pathology, where several subjects had negative 18F-AV-1451 despite positive CSF biomarkers." How can a patient be classified as stage IV based on PET, while having no PET-signal at all? Doesn't stage VI mean widespread tau pathology? Did the authors mean Abeta-negative?

Fig 7: How many patients fall into the different categories on the x axis?

Referee #3 (Remarks):

This is a timely study addressing an important area that is the subject of intense academic interest. The data are novel and the authors are well recognized experts in this area.

Specific comments:

1. In Figs 2 - 4 associations are adjusted for diagnostic group membership. As the authors and others have shown, AV1451 uptake is highly correlated with cognitive performance. Diagnostic group membership in turn is essentially a surrogate for cognitive performance. The result is that adjusting out the effect of group membership on the correlation in Fig 2 in large measure adjusts out the effect of AV1451. In Fig 3, I realize the AV1451 vs GM map is convincing (Fig a) but by adjusting for diagnostic group, an association between tau and GM is greatly reduced. In Fig 4 MMSE, Delayed recall, AQT are in large measure equivalent to diagnostic group, so adjusting away the effect of group removes variance attributable to cognition. My sense is that if the data in Figs 2-4 were not adjusted for diagnostic group membership, the conclusions would be different.
2. Fig 1. Doing correlations within each clinical group is the primary analysis, but this truncates the range of values thereby limiting correlations. I would think that the primary analysis should pool all 3 clinical groups, and correlate T tau and P tau with AV1451 in the merged stage 1-5 AV1451 ROIs. This would give correlations over the full range of tau and the full range of AV1451 values. This is mentioned in the discussion, but I would think pooling the 3 clinical groups would be the primary analysis.
3. The authors claim that AV1451 is not elevated in AB+ vs AB- controls, but this contradicts existing literature on this subject. How do the authors reconcile this contradiction?
4. The point above leads to a conclusion that appears to be a difficult to rationalize. Fig 6 shows higher tau in AB+ than AB- controls. The literature from other groups is clear on the fact that AB+ controls have more AV1451 uptake than AB- controls. Based on the principle of transitivity would therefore expect CSF tau and AV1451 to be correlated in controls (assuming AB+ and AB- were pooled).
5. The authors found normal CSF tau in some AD dementia cases. Don't these cases suggest that either the diagnosis is wrong or the CSF tau measurements were in error? While tau may plateau, is it reasonable to conclude that tau levels return fully to normal in AD dementia cases?
6. One aspect of the discussion that is missing is this. While P tau and T tau are state rather than stage markers (an important point to make), but they are not markers of the same pathological states. T tau is elevated in AD but also in head trauma, stroke, and CJD while P tau is only elevated in AD. The evidence seems overwhelming therefore that T tau and P tau cannot indicate the same pathological state. Logically T tau represents neuronal injury which occurs in AD but also other conditions while P tau is a marker of the state leading to AD tau.

Referee #1 (Comments on Novelty/Model System):

The authors have designed their study with an a priori knowledge on Braak stages and CSF Tau biomarkers. They should justify this approach since there is no evidence that Tau PET ligand or CSF Tau biomarkers reflect neurofibrillary degeneration. Correlation/Covariance of raw data may have also been of interest without any a priori approach.

- We focused on a number of a priori specified regions to reduce the number of statistical comparisons, and reduce the likelihood of false positive findings. We also provide results from voxelwise regression analyses that do not depend on a priori defined stages/regions. These are presented in Figure 2 and mentioned in the text (page 6, lines 3-7).
- We acknowledge that more work is needed to validate both CSF tau biomarkers and tau PET tracers against neuropathology. Few studies have compared CSF tau biomarkers with neuropathology, and the results have differed somewhat (page 13, lines 5-10). The few neuropathology studies that have been conducted to evaluate AV-1451 have found strong correlations with AD-type tau pathology, although off-target binding has been observed and is yet to be explored further, an issue which we now discuss (page 14, lines 5-11).

Referee #1 (Remarks):

First, I must apologize for the delay in my review.

Mattsson and coll. have investigated the association between CSF Tau biomarkers and the Tau PET ligand AV1451 using linear regression models. They also analyzed associations between Tau markers (CSF and PET) and atrophy (VBM approaches) and between Tau markers and cognition (MMSE, ADAS-Cog).

This is an original study, well written and well done. Nevertheless, it needs some clarifications I have only minor comments:

1) First, it will be interesting to see any association without any a priori approach (w/o Braak stages, w/o cut-offs...) as in Suppl Fig1. The sample population is low and such stratification may lower statistical power but the choice of the authors has to be justified.

- Voxelwise correlations that do not depend on stages/regional segmentation are presented in Figure 2, and these analyses has been performed without any “any a priori approach” (page 6, lines 3-7).

2) The positive PET signal seen in Fig 3 and Suppl. in the locus coeruleus is not included in any analyses. Since it was reported that Tau pathology starts in this area, the authors may include a Braak stage 0 to confirm (or not) this information.

- The positive signal in the brain stem in Fig 3 and Supp Fig 2 is for associations between CSF tau and VBM. The signal suggests that there may be a weak correlation between greater CSF tau levels and atrophy in the brain stem. In contrast, there was no sign of correlation between CSF tau measures and tau PET in the brain stem (as seen in Fig 2). The locus coeruleus is too small a structure to be accurately analyzed by PET (resolution 4-5 mm, which is greater than the average diameter of locus coeruleus) without substantially underestimating the regional signal.

3) The discussion needs to be more balanced: such Tau PET ligands are not fully specific and other targets have been identified.

- We acknowledge that there may be issues regarding specificity and sensitivity of tau PET in relation to different neuropathological changes. One post mortem brain tissue study found that AV-1451 bound strongly to tau lesions consisting of paired helical filaments in Alzheimer's disease, but not to other tau inclusions (Marquie et al, Ann Neurol 2015). On the other hand, AV-1451 also bound partly to neuromelanin- and melanin-containing cells, and to brain hemorrhagic lesions. Off-target binding is known to be present for example in the brainstem, in the choroid plexus and in the basal ganglia (Smith et al, Mov Dis 2017).

This is one of the reasons why we did not measure AV-1451 in the hippocampal formation, which is close to the choroid plexus, and thereby prone to artifacts from off-target binding. We have expanded the discussion about this (page 14, lines 5-11).

4) In AD dementia, there is a subset of patients with high Tau concentrations (as suggested by SD in Table 1). Do they have a particular Tau PET distribution?

- All individual CSF tau data is shown in Figure 1. There was indeed a number of AD dementia patients with quite high CSF tau concentrations (for example, n=8 for CSF T-tau > 1000 ng/L). These patients do not appear to differ from other AD dementia patients in AV1451 PET signal in stage I-II regions (panels A and G), but they do appear to have higher AV1451 signal in stage III, stage IV, stage V and stage VI regions (panels B-E, H-K). This explains the positive correlation between CSF tau and AV1451 in AD dementia in those regions.

5) In figure 1, is each spot an individual person, a combination of the three CSF quantification? Why the merged graph does not show more values?

- Yes, in figure 1 each spot is an individual person. We now clarify this in the figure legend. T-tau is shown in the top row and P-tau in the bottom row. The merged panels (F and L) show data for tau PET in a composite of tau stage I-V regions.

Referee #2 (Remarks):

The manuscript is extremely hard to read and therefore also difficult to review properly. After spending several hours for reading it twice, I still don't fully comprehend figures 6 and 7. I think, in the current presentation the manuscript is only readable for the group of authors and their immediate competitors. The subsections headings, figure title and even the manuscript title give absolutely no clue about the findings. There is not even a single verb in any of those headings. The result section is particularly difficult to read. The subsections don't start with a rationale and don't finish with a conclusion. In the legends the individual panels should be mentioned and described specifically. Nevertheless, the manuscript contains some interesting data on the time course of biomarkers in AD, which are largely hidden due to poor presentation.

- We apologize for the unclear writing. We have revised the title, the figure titles, the figure legends, the subsections heading, the results, and the discussion.

The authors seem to misuse the word "disease state markers", which normally refers to a biomarker that is positive only in a certain disease stage. From the data it seems that both CSF-tau and PET-tau are disease state markers that become positive in different disease stages. If at all, the common terminology trait vs. state marker should be used.

- We adhere to the definitions of "state" and "stage" biomarkers presented in a Lancet Neurology paper on AD biomarkers (Blennow K, Hampel H. Lancet Neurol 2003; 2: 605–13). By this definition, a state marker reflects the intensity of the disease process. One example is CSF T-tau, which is increased in proportion to the intensity of ongoing neuronal damage, and thus is much higher in e.g. Creutzfeldt-Jakob disease than in AD. In contrast, the corresponding stage marker gives a measure of how far the degenerative process has proceeded. The stage marker for neurodegeneration in disorders such as AD and CJD is CT or MRI measures of atrophy. Following this line of reasoning, tau PET is a stage marker of tau pathology. This is similar to how these definitions are used in other areas of medicine. For example, in arthritis, inflammatory proteins in blood serve as state markers, with X-ray measures of degree of cartilage destructions in the joints is the corresponding stage marker. Similarly, increased blood glucose is a state marker for diabetes. The term trait marker (a *behavioral* characteristic induced by the expression of one or more genes) is often used in psychiatry, but does not fit in this context, also since a trait exists not only during, but *also before*, the disease onset. We present a rationale in the discussion (page 11, lines 1-9).

The abbreviation "AD-ctx" is also confusing - why not call it atrophy? Does "T-tau ($\beta=0.00024$ [unit: ng/L], $p=0.015$) and P-tau ($\beta=0.0025$ [unit: ng/L], $p=0.0021$) had positive associations with AD-cortex in these adjusted models" mean more Tau is associated with thicker or thinner cortex? In the discussion the answer is finally revealed.

- We used two proxies of atrophy: hippocampal volume and cortical thickness in a region of interest associated with AD ("AD-cortex"). We used the term AD-cortex instead of "atrophy" to make it clear which measure we are discussing, since results sometimes differed between AD-cortex and hippocampal volume. We have revised the language to make the associations between tau and thicker/thinner cortex clearer (page 6, lines 18-20).

Tau staging should be explained in the result section or maybe even indicated in the figure 1 or 2. Do the results from Fig. 1 match with the voxel-based result from Fig 2?

- We have now explained the tau staging in the results section (page 5, lines 8-18). The regional results in Figure 1 generally agreed with the voxelwise results in Figure 2 (page 6, line 6).

Does stage IV mean patients with Stage IV PET pattern or does it denote the region that defines stage IV patients. I assume the latter is correct and "region IV" would be a less misleading term.

- The term "stage IV" is used to describe a brain region of interest (specifically, the inferior temporal and middle temporal cortex). This terminology was introduced by Cho et al (Annals of Neurology, 2016), who also defined the regions for each stage. Since we use the Cho terminology, we prefer to keep the term "stage". But we have clarified that this is an image-based tau staging (page 5, lines 9-10). Note that we did not assign a single "stage" to individual patients (page 19, lines 15-16). Instead, we measured the AV-1451 signal within different regions of the brain, labelled the stage I-II region, the stage III region, the stage IV region, the stage V region, and the stage VI region. Each subject therefore had data for all of the tau stage regions.

"Tau biomarkers and preclinical AD": The authors should define "Abeta status" in result section. In that context it could mean PiB positive or CSF-marker positive. In fact, including amyloid PET in the correlation analysis would be nice.

- We have clarified that CSF AB42 was used to define Abeta status (page 7, lines 17-18). The findings were the same when defining A β -positivity by 18F-flutemetamol PET A β -imaging instead (page 7, lines 23ff).

"Sensitivity of tau biomarkers for AD"

Are the figures mislabeled? The text mentions stages and dichotomous tau measures, but the figure shows continuous tau measures and nothing related to 18F-AV-1451 stages. How were the cut-offs determined? I didn't understand that part at all.

- This section refers to Figure 6, which has two panels with sensitivity for different tau measures to detect AD. Panel A shows the sensitivity for 18F-AV-1451 in different tau stage regions. For example, it shows that a positive signal in the stage III region had 13% sensitivity for preclinical AD, 86% sensitivity for prodromal AD, and 100% sensitivity for AD dementia. Panel B shows the sensitivity for 18F-AV-1451 in the stage I-V composite region, CSF T-tau and P-tau. For example, it shows that CSF T-tau positivity had 40% sensitivity for preclinical AD, 71% for prodromal AD and 80% for AD dementia. The cut-offs were determined in the Abeta-negative controls, using their mean levels plus two standard deviations, as described (page 8, lines 8-10). We have rewritten the section about sensitivity to make it clearer (page 8, line 7ff).

Page 8: "isolated CSF tau positivity was most common for stage VI tau pathology, where several subjects had negative 18F-AV-1451 despite positive CSF biomarkers." How can a patient be classified as stage IV based on PET, while having no PET-signal at all? Doesn't stage VI mean widespread tau pathology? Did the authors mean Abeta-negative?

- We have rewritten this section, which describes the concordance between CSF tau measures and regional 18F-AV-1451 measures (page 9, line 9ff). Note that each person was classified as positive/negative for CSF T-tau, P-tau, and 18F-AV-1451 in each of the tau stage I-II region, stage III region, stage IV region, stage V region, stage V region I, and the stage I-V composite region. For example, an AD subject with early stage AD may be positive in PET-regions I-II and III, but negative in regions IV-VI and have pathological CSF T-tau. This person will then be classified as negative for 18F-AV-1451 in the stage IV region, since the 18F-AV-1451 signal in the stage IV region was below the cutoff (this does not mean that there was **no** PET signal in the stage IV region).

Fig 7: How many patients fall into the different categories on the x axis?

- Every patient is represented in each category on the x-axis. Each patient is classified as positive or negative for 18F-AV-1451 in the tau stage I-II region, the tau stage III region, the tau stage IV region, the tau stage V region, the tau stage VI region and the tau stage I-V composite region. (page 26 lines 18-20)

Referee #3 (Remarks):

This is a timely study addressing an important area that is the subject of intense academic interest. The data are novel and the authors are well recognized experts in this area.

Specific comments:

1. In Figs 2 - 4 associations are adjusted for diagnostic group membership. As the authors and others have shown, AV1451 uptake is highly correlated with cognitive performance. Diagnostic group membership in turn is essentially a surrogate for cognitive performance. The result is that adjusting out the effect of group membership on the correlation in Fig 2 in large measure adjusts out the effect of AV1451. In Fig 3, I realize the AV1451 vs GM map is convincing (Fig a) but by adjusting for diagnostic group, an association between tau and GM is greatly reduced. In Fig 4 MMSE, Delayed recall, AQT are in large measure equivalent to diagnostic group, so adjusting away the effect of group removes variance attributable to cognition. My sense is that if the data in Figs 2-4 were not adjusted for diagnostic group membership, the conclusions would be different.

- We agree that diagnostic group is correlated with both atrophy and with cognition. This is the reason why we adjusted for diagnostic group. Without this adjustment, some correlations between tau measures and atrophy and cognition would be stronger, but one could then argue that the associations were confounded by diagnoses. By adjusting for diagnosis, we believe that we provide more robust estimates of the correlations between tau and atrophy and cognition. (page 12, lines 4-13)

2. Fig 1. Doing correlations within each clinical group is the primary analysis, but this truncates the range of values thereby limiting correlations. I would think that the primary analysis should pool all 3 clinical groups, and correlate T tau and P tau with AV1451 in the merged stage 1-5 AV1451 ROIs. This would give correlations over the full range of tau and the full range of AV1451 values. This is mentioned in the discussion, but I would think pooling the 3 clinical groups would be the primary analysis.

- We apologize for being unclear in the methods, and we have now made the text clearer (page 20, lines 3-6). The analysis was done on the pooled group, but with diagnosis entered as a covariate. This made it possible to extract correlations for each diagnostic group. As we explain above, we believe that adjusting for diagnosis is necessary to avoid confounding effects of diagnosis on the tested associations.

3. The authors claim that AV1451 is not elevated in AB+ vs AB- controls, but this contradicts existing literature on this subject. How do the authors reconcile this contradiction?

- AV-1451 was not significantly elevated in AB+ vs AB- controls in this study, as shown in Supplementary Figure 3, (stage I-II, $P=0.53$; stage III, $P=.19$; stage IV, $P=0.63$; stage V, $P=0.30$; stage VI, $P=0.72$; stage I-V, $P=0.31$). The significant effects on CSF tau

biomarkers, but not on AV-1451, suggest that CSF tau biomarkers may change before AV-1451 in preclinical AD. Although effects on CSF tau dominated, we note that the lack of significant effects on AV-1451 could partly be a power issue, since the mean levels tended to be higher in AB+ controls (page 7, lines 21-23). One recent large study (N=490) found correlations between AV1451 and PiB-PET in cognitively healthy controls (Vemuri et al, *Alz Dem* 2017). Previous smaller studies (e.g. Schöll et al, *Neuron* 2016) have also found correlations between AV-1451 and AB+ in cognitively healthy elderly, so there may be cohort-dependent differences that are not yet fully understood. (page 14, lines 1-5)

4. The point above leads to a conclusion that appears to be difficult to rationalize. Fig 6 shows higher tau in AB+ than AB- controls. The literature from other groups is clear on the fact that AB+ controls have more AV1451 uptake than AB- controls. Based on the principle of transitivity would therefore expect CSF tau and AV1451 to be correlated in controls (assuming AB+ and AB- were pooled).

- The correlations between AV1451 and CSF tau were not significant in the controls. The correlations for the controls that are shown in Figure 1 is from the pooled data (with coefficients extracted for the controls). We have also tested when restricting the models only to the controls, again with no significant results. We can therefore conclude that we do not find any strong correlations between AV1451 and CSF tau in the controls included in this study. Potentially, this is a sample size issue, if there is a slight correlation which we were not powered to detect.

5. The authors found normal CSF tau in some AD dementia cases. Don't these cases suggest that either the diagnosis is wrong or the CSF tau measurements were in error? While tau may plateau, is it reasonable to conclude that tau levels return fully to normal in AD dementia cases?

- It is true that several AD dementia patients had normal CSF T-tau and P-tau levels. But we do not think that CSF tau measures can be used alone to exclude AD, especially in people who have evidence of amyloid-pathology. It is known from previous studies that CSF tau levels vary among AD patients. In our clinical practice, we have used CSF tau biomarkers for about 20 years, and we sometimes see AD dementia patients with normal CSF tau despite typical clinical symptoms and normal disease progression. As shown here, such patients also have biomarker evidence of amyloid pathology, and have tau pathology as measured by AV1451. In our opinion, this makes it very unlikely that they are misdiagnosed. Nevertheless, the variations in CSF tau levels may have a clinical relevance, since CSF tau levels may be related to rate of disease progression (page 11, lines 22ff). It may therefore be interesting to stratify AD dementia patients by their CSF tau levels. Unfortunately, we cannot do that in the present paper, since that would introduce a bias in any comparisons with AV-1451.

6. One aspect of the discussion that is missing is this. While P tau and T tau are state rather than stage markers (an important point to make), but they are not markers of the same pathological states. T tau is elevated in AD but also in head trauma, stroke, and CJD while P tau is only elevated in AD. The evidence seems overwhelming therefore that T tau and P tau cannot indicate the same pathological state. Logically T tau represents neuronal injury which occurs in AD but also other conditions while P tau is a marker of the state leading to AD tau.

- We are grateful for this comment and we have expanded the discussion about this (page 13, lines 1-5).

Thank you for the submission of your revised manuscript to EMBO Molecular Medicine. We have now received the enclosed reports from the referees that were asked to re-assess it. As you will see the reviewers are now globally supportive and I am pleased to inform you that we will be able to accept your manuscript pending the following final amendments:

Please address (in writing) the minor comments made by referee 1, and include a discussion for point 2) in the main text. Please provide a letter INCLUDING the reviewer's reports and your detailed responses to their comments (as Word file).

Please submit your revised manuscript within two weeks. I look forward to seeing a revised form of your manuscript as soon as possible.

***** Reviewer's comments *****

Referee #2 (Remarks):

The manuscript significantly improved.

Referee #3 (Remarks):

This is an important but complex paper. The authors have done a good job revising in response to reviewer comments. I have just a couple things to add.

1. I still disagree with including clinical diagnosis as a covariate when assessing correlations among CSF tau, AV1451, atrophy and cognition. The authors rationale is "associations would confounded by diagnoses". Bu diagnosis is not a confounder to cognitive performance -- the two are equivalent (diagnosis is simply a crude measure of cognitive performance). Why is someone labeled dementia vs normal? - because cognitive performance is bad in the former and not in the latter.

2. This is something I should have mentioned on the first review but forgot to - my apologies. The interpretation of CSF tau as a state measure (that becomes abnormal early and does not change greatly as the disease progresses) conflicts with interpretation in a well-known paper by Buchhave et al Arch Gen Psych 2012. In the Buchhave paper CSF Ptau and T tau became progressively worse the closer individuals were to dementia - the interpretation was that CSF tau does change as the disease progresses (whereas CSF Ab does not). This seems to contradict the authors' current interpretation of CSF tau - and 3 of the authors of the current paper were also authors of the Buchhave paper. Given how well known the Buchhave paper is, it might be worth an explanation in this paper.

2nd Revision - authors' response

14 June 2017

Reviewer comments:

Referee #2 (Remarks):

The manuscript significantly improved.

* We are grateful for these kind words.

Referee #3 (Remarks):

This is an important but complex paper. The authors have done a good job revising in response to reviewer comments. I have just a couple things to add.

1. I still disagree with including clinical diagnosis as a covariate when assessing correlations among CSF tau, AV1451, atrophy and cognition. The authors rationale is "associations would confounded by diagnoses". Bu diagnosis is not a confounder to cognitive performance -- the two are equivalent (diagnosis is simply a crude measure of cognitive performance). Why is someone labeled dementia vs normal? - because cognitive performance is bad in the former and not in the latter.

- We believe that diagnosis of AD is a confounder in a statistical sense, since it is related both to the tau measures (the model predictors) and atrophy/cognitive decline (the model outcomes). In our opinion, the fact that there were significant correlations between more AV-1451 and worse atrophy/cognitive decline even when adjusting for diagnosis supports the theory that AV-1451 has properties of a disease stage biomarker, which may change dynamically as the disease progresses, even in people with symptomatic AD. In contrast, the fact that CSF tau measures were not correlated with atrophy/cognitive decline in models adjusted for diagnosis, supports the theory that CSF tau is a disease state biomarker, which does not change much in people with symptomatic AD, despite that atrophy/cognitive decline continues.

2. This is something I should have mentioned on the first review but forgot to - my apologies. The interpretation of CSF tau as a state measure (that becomes abnormal early and does not change greatly as the disease progresses) conflicts with interpretation in a well-known paper by Buchhave et al Arch Gen Psych 2012. In the Buchhave paper CSF Ptau and T tau became progressively worse the closer individuals were to dementia - the interpretation was that CSF tau does change as the disease progresses (whereas CSF Ab does not). This seems to contradict the authors' current interpretation of CSF tau - and 3 of the authors of the current paper were also authors of the Buchhave paper. Given how well known the Buchhave paper is, it might be worth an explanation in this paper.

- We are grateful to the reviewer for pointing this out. There are a few different possible interpretations of the findings in the Buchhave paper, which was based on cross-sectional CSF tau data and longitudinal clinical data. The main finding was that people with MCI who converted to AD dementia within a few years had higher CSF tau levels than people who converted to AD dementia later. One possible interpretation of these results, as highlighted by the reviewer, was that CSF tau increased as people approached the dementia stage. However, since the study only included cross-sectional CSF tau data it did not test if these measures changed over time. An alternative interpretation of the results was that differences in CSF tau reflected differences in the intensity of the neurodegenerative process. With this interpretation, MCI patients with high CSF tau had a more rapidly progressive form of AD compared to patients with low CSF tau. People with high CSF tau would then be more likely to develop AD dementia sooner. This view is supported by studies showing that high CSF tau predicts a more rapid clinical deterioration (Wallin et al, Neurology 2010). We now comment on this in the discussion section (page 13, lines 1-7).

Corresponding Author Name: Niklas Mattsson
Journal Submitted to: EMBO Molecular Medicine
Manuscript Number: